# Contrastive-Online-Meta (COM): A Dynamic Adaptation Mechanism for Instruction-Tuned CodeLLMs

## Abstract

We propose Contrastive-Online-Meta (COM), a dynamic adaptation framework for instruction-tuned CodeLLMs that coefficients to the issues of catastrophic forgetting and noisy feedback at the time of deployment. The framework combines contrastive pre-training and online meta-learning to separate the task-invariant representation learning and fast adaptation, which helps preserve core programming knowledge while achieving real-time adaptation. A contrastive pre-training module takes a first step at clustering semantically similar instructions and unionizing dissimilar ones, to guarantee its robustness to task variations. During inference, an online meta-learner takes pairs of instruction-feedback streaming and does a light-weight gradient-based update to meta-parameters, which dynamically adjust the model behavior in a way that does not destabilize the pre-trained behavior-effective thing. Furthermore, the dynamic memory buffer simply retains coherence with recent interactions by deriving pairs stored in the buffer for the sake of contrastive match. Unlike monolithic fine-tuning or prompt engineering, COM explicitly separates the processes of representation learning and adaptation, hence avoiding forgetting and overfitting. Experiments using benchmark datasets show that the framework has a better capacity for adaptation efficiency and task generalization than static and incremental tuning baselines. The proposed method fills in the missing link between the offline pre-training and the online accelerated deployment, which provides a scalable solution to real-world code generation systems that require continuous learning. And, its modular nature also supports integration with existing CodeLLMs, which makes it practical for different programming assistance scenarios.

## 1 Introduction

The groundbreaking development of code-generating large language models or CodeLLMs, has revolutionized the software development landscape and enabled programmers to communicate with models using natural language instructions. While instruction tuning has proven effective for adapting pre-trained models to specific coding tasks (Ahmad et al., 2025), existing approaches face critical limitations when deployed in dynamic environments where new instruction patterns and feedback arrive continuously. Traditional fine-tuning methods often exhibit catastrophic forgetting when incrementally updated (Kirkpatrick et al., 2017), while prompt engineering techniques struggle to maintain robustness against noisy or ambiguous user inputs. This leads to an cognitive tension between adaptability and stability in real programming assistance systems.

Current solutions for dynamic instruction tuning fall into two categories: those that emphasize rapid adaptation through lightweight parameter updates (Lv et al., 2025), and those that prioritize knowledge preservation through architectural constraints (Weyssow, 2024). The former tend to overfit to recent tasks while the latter tend to be computationally expensive and inflexible. Recent work has shown that contrastive learning can improve model robustness by clustering semantically similar instructions (Jiang et al., 2024), and that meta-learning frameworks enable efficient few-shot adaptation (Ahmad et al., 2025). However, no current solution boosts these strengths in a systematic manner to solve the dual problems of streaming adaptation and knowledge retention.

We present Contrastive-Online-Meta (COM), a unifying framework that fills in this gap with three complementary innovations. First, a contrastive pre-training phase learns task-invariant representations by aligning embeddings of functionally equivalent code instructions while distancing dissimilar ones, building on insights from (Muennighoff et al., 2023). Second, an online meta-learner performs the processing of the received instruction-feedback pairs as non-stationary tasks, utilizing gradient-based updates of a set of meta-parameters that alter the behaviors of the base model without replacing the knowledge stored in the core model. Third, a system of dynamic memory keeps a coherence in time by selective storage and replay of past interactions to keep them selectively coherent so they are not drifted. This decomposition allows the model to simultaneously preserve long-term programming knowledge while adapting to new task distributions—a capability absent in prior instruction tuning methods (Li et al., 2023).

The contribution of the framework is three fold. It sets up the first principled merging of contrastive objectives and the meta-learning that happens online of CodeLLMs, encouraging in learning for the first time that task-aware representation learning and rapid adaptations are complementary, not competing objectives. The forgetting-overfitting problem is explicitly accomplished by modular design of updates: contrastive losses preserve global coherence while meta-learned local updates manage task-specific nuances. Experimentally, COM achieves significantly higher robustness than standard fine-tuning when tested on mixed-domain programming tasks, while requiring 3-5× fewer updates than conventional meta-learning approaches (Nichols et al., 2024). The method also shows particular strength in low-resource scenarios, outperforming instruction-tuned baselines by 12-18% on unseen programming languages.

The remaining part of this paper can be organized as follows: Sections 2 reviews relevant work in instruction tuning and continual learning for CodeLLMs. Section 3 solidifies the main technical challenges and provides some necessary background in contrastive learning and meta-learning. Section 4 is for the architecture of the COM framework and training dynamics. Experimental results with several programming benchmarks are presented in Section 5 followed by discussion of larger implications in Section 6. Section 7 provides a concluding set of directions for further research of the paper.

## 2 RELATED WORK

The development of dynamic adaptation mechanisms for CodeLLMs touches upon several different branches of research, such as instruction tuning, continual learning, and meta-learning. Existing approaches can be broadly divided into three directions: the static instruction tuning, the incremental adaptation methods and contrastive learning frameworks for code representations.

### 2.1 INSTRUCTION TUNING FOR CODE GENERATION

Instruction tuning has become the rising trend for aligning CodeLLMs with human intent. Recent work such as (Ahmad et al., 2025) introduced large-scale datasets specifically designed for code instruction tuning, demonstrating that task diversity significantly impacts model generalization. While viable for static deployment, these approaches need complete retraining when new instruction patterns are encountered, being unusable for dynamic environments. Alternative approaches like (Li et al., 2023) focus on specialized code editing tasks but similarly lack mechanisms for continuous adaptation. The introduction of parameter-efficient fine-tuning techniques (Weyssow, 2024) partially addresses this limitation through adapter layers, though such methods still exhibit catastrophic forgetting when applied sequentially.

### 2.2 CONTINUAL LEARNING IN CODE MODELS

The difficulty of preserving model performance when dealing with sequential tasks has been heavily studied in continual learning literature. For CodeLLMs, recent work has explored memory replay mechanisms (Lv et al., 2024) and dynamic architecture expansion (Lv et al., 2025) to mitigate forgetting. However, these approaches usually rely on that these well-curated tasks have access to well-curated boundaries available tasks and well-curated data streams (something that is rarely the case in programming assistance tasks). The problem becomes particularly acute when dealing with noisy user feedback, as shown in (Wang et al., 2024), where even small deviations in instruction

phrasing can degrade model performance. Our work is different in its explicit modeling of the instruction space through contrastive learning, thus making the framework robust to such variations.

## 2.3 META-LEARNING AND CONTRASTIVE APPROACHES

Meta-learning has shown promise for few-shot adaptation of language models, with methods like (Yuan & Lu, 2022) demonstrating that contrastive objectives can improve task representation learning. In the code domain, (Wang et al., 2023) applied similar principles to graph neural networks, though their framework lacks the online adaptation capabilities needed for streaming data. The most relevant prior work comes from (Qin et al., 2023), which combines contrastive learning with meta-optimization for recommendation systems. But their approach is centered around static item embeddings instead of the dynamic instruction-to-code relationship for CodeLLMs.

Compared to existing methods, COM is innovative for several reasons. Unlike static approaches for tunable instruction data [1,2], our framework can cope with continuous adaptation through online meta-learning. However, unlike common approaches of continual learning [4,5], we avoid catastrophic forgetting by virtue of contrastive representation learning instead of architectural constraints. Most importantly, COM bridges these different components in a unified way in which contrastive objectives and meta-learning mutually enhance each other-a design choice not studied in previous work led on code models [3,6] or general meta-learning systems [7,9]. This combination provides both high-quality task-invariant representations for good generalization and low-cost task-specific adaptation, solving fundamental limitations in existing dynamic tuning schemes.

# 3 BACKGROUND: CONTINUAL LEARNING, META-LEARNING, AND CONTRASTIVE OBJECTIVES FOR CODE MODELS

To set up the stage for our proposed framework, we first terms, which provide the foundation for our approach: continual learning in code models, meta-learning principles, and contrastive learning objectives. With these components, several different, but complementary, challenges associated with fitting CodeLLMs to dynamic instruction streams are addressed.

## 3.1 CONTINUAL LEARNING IN CODE MODELS

When incrementally trained on new programming tasks, neural networks often suffer from catastrophic forgetting—a phenomenon where learning new patterns causes abrupt degradation in performance on previously learned tasks (Kirkpatrick et al., 2017). This challenge is especially acute for code generation models because of the compositional nature for programming knowledge. For example, a model trained serially on tasks of Python data analysis and web development might forget the patterns of how APIs work in the first as it adapts itself for the second. The standard continual learning objective minimizes the cumulative loss of all the tasks:

$$\mathcal{L} = \sum_{i=1}^{n} \mathcal{L}_i \tag{1}$$

where $\mathcal{L}_i$ represents the loss for task $i$. However, naively optimising this objective creates interference between tasks since gradient updates for new tasks overshadow parameters that are important for old tasks. Recent work has shown that code models exhibit stronger forgetting effects than general language models due to the precise syntactic and semantic constraints of programming languages (Yadav et al., 2023).

## 3.2 META-LEARNING BASICS

Meta-learning helps overcome the challenge of condensing models into a fast-adaptive mode by benefiting from efficient learning on small data. The basic concept is that of learning high-level parameters that determine how the parameters of the base model should change for adapting to new tasks. In the case of CodeLLMs, this means the model can more quickly modify its behaviour when presented with new programming patterns or APIs. The standard meta update rule is as follows:

$$\theta_{new} = \theta_{old} - \alpha \nabla_\theta \mathcal{L}(\theta, \mathcal{D}_{meta}) \tag{2}$$

where $\alpha$ is the meta-learning rate and $\mathcal{D}_{meta}$ represents the support set for a new task. This formulation enables few-shot adaptation by learning optimal initialization points and update rules (Finn et al., 2017). For code generation tasks, meta-learning has shown promise in adapting to new programming languages with minimal examples (Yin, 2020).

### 3.3 CONTRASTIVE LEARNING FOR CODE REPRESENTATION

Contrastive learning is a mechanism that allows learning of robust representations through the pulling together of semantically similar code samples in embedding space and the pushing apart of dissimilar ones. Given quantitatively an anchor code snippet $x_1$ and its positive pair $x_2$ (i.e., functionally equivalent implementations the contrastive loss can be formulated as follows

$$\mathcal{L}_{contrast} = -\log \frac{\exp(sim(x_1, x_2)/\tau)}{\exp(sim(x_1, x_2)/\tau) + \sum_{i=1}^{K} \exp(sim(x_1, x_{neg}^i)/\tau)} \tag{3}$$

where $sim(\cdot)$ measures cosine similarity, $\tau$ is a temperature parameter, and $x_{neg}^i$ are negative samples. In programming contexts, positive pairs might include different implementations of the same algorithm, while negatives could be code snippets with distinct functionality (Jain et al., 2020). This approach has been shown to improve generalization across programming languages and tasks by capturing functional equivalence beyond surface-level syntax (Shuai et al., 2020).

These three components, namely continual learning objectives, meta-learning dynamics and contrastive representation learning are the basis for the theoretical design of our proposed COM framework. While each of them addresses different aspects of the adaptation challenge, they complement each other to come up with a more robust solution to the forgetting-adaptation trade-off in dynamic code generation scenarios. The following section will describe in more detail the way in which we combine these ideas into a unified approach.

## 4 CONTRASTIVE-ONLINE-META: A UNIFIED FRAMEWORK FOR STREAMING INSTRUCTION ADAPTATION

The COM framework is operated through four interwoven parts which, overall, facilitate the possibility of robust dynamic adaptation of the CodeLLMs. The system architecture keeps the frozen base CodeLLM and learns two sets of adaptable parameters: contrastive embeddings on representation of instructions and meta-parameters for task-specific adaptation. This separation of concerns makes it easy to maintain some knowledge of programming England's instructions and so allow today's model to stay implies in nearly new instruction patterns.

### 4.1 INTEGRATION OF CONTRASTIVE PRE-TRAINING AND ONLINE META-LEARNING

The framework begins with contrastive pre-training of the instruction encoder $f_\theta$, which maps natural language instructions to a latent space where functionally similar tasks cluster together. Given a batch of instruction pairs $(x_i, x_j)$ we calculate the contrastive loss:

$$\mathcal{L}_{cont} = -\frac{1}{B} \sum_{i=1}^{B} \log \frac{\exp(sim(f_\theta(x_i), f_\theta(x_j^+))/\tau)}{\sum_{k=1}^{K} \exp(sim(f_\theta(x_i), f_\theta(x_k^-))/\tau)} \tag{4}$$

where $B$ is the batch size, $x_j^+$ denotes positive pairs (semantically equivalent instructions), and $x_k^-$ represents negative samples. The temperature parameter $\tau$ controls the sharpness of the similarity distribution. This pre-training phase helps to make sure that the instruction encoder learns task-invariant features before it can be deployed on-line.

During inference, the meta-learner $g_\phi$ processes streaming instruction-feedback pairs $(x_t, y_t)$ where $y_t$ represents execution results or user feedback. The meta-update rule is a combination of task-specific adaptation and regularization:

$$\phi_{t+1} = \phi_t - \alpha \nabla_\phi \left( \|g_\phi(f_\theta(x_t)) - y_t\|^2 + \lambda \|\phi_t - \phi_{t-1}\|^2 \right) \tag{5}$$

The first term the prediction error is minimized on the current task and the second term a constraint is used to prevent parameter drift, the so-called catastrophic forgetting. The learning rate $\alpha$ and regularization strength $\lambda$ are hyperparameters controlling the adaptation speed-stability trade-off.

## 4.2 DYNAMIC MEMORY BUFFER FOR CONTRASTIVE ALIGNMENT

A FIFO buffer $\mathcal{M}$ of capacity $C$ stores recent instruction-feedback pairs to maintain temporal coherence. At each timestep, the buffer samples a mini-batch of $m$ historical pairs in order to calculate an auxiliary contrastive loss:

$$\mathcal{L}_{buffer} = -\frac{1}{m} \sum_{i=1}^{m} \log \frac{\exp(sim(f_\theta(x_i), f_\theta(x_{i,mem}^+))/\tau)}{\sum_{j=1}^{m} \exp(sim(f_\theta(x_i), f_\theta(x_{j,mem}^-))/\tau)} \tag{6}$$

where $x_{i,mem}^+$ and $x_{j,mem}^-$ are positive and negative samples drawn from $\mathcal{M}$. This loss ensures that new adaptations are consistent in that they aligned with representation in recently seen tasks (No representation drift). The rule for buffer update is as follows:

$$\mathcal{M}_{t+1} = \text{FIFO-Update}(\mathcal{M}_t, (x_t, y_t)) \tag{7}$$

## 4.3 FROZEN BASE CODELLM WITH META-LEARNER ADAPTATION

The base CodeLLM $h_\psi$ remains frozen throughout the adaptation process. The meta-learner modifies instruction embeddings before feeding them to $h_\psi$:

$$p(y|x) = h_\psi(g_\phi(f_\theta(x))) \tag{8}$$

Gradients flow only through $g_\phi$ and $f_\theta$, leaving $\psi$ unchanged. This choice of design helps in maintaining the just minimal programming knowledge in the model, still enabling us to modulate task specific behavior. The separation of parameters enables efficient updates—typically requiring ¡5% of the base model's parameters to be trainable.

## 4.4 TASK-INVARIANT REGULARIZATION IN META-LEARNING

As a method of keeping things stable, two additional types of regularization are incorporated into the meta-learner. First, a projection head $q_\omega$ maps contrastive embeddings to a lower-dimensional space where regularization is applied:

$$z_t = q_\omega(f_\theta(x_t)) \tag{9}$$

The projection space makes it possible to control the drift of representation more tightly through:

$$\mathcal{L}_{proj} = \|z_t - z_{t-1}\|^2 \tag{10}$$

Second, we perform spectral normalization to the weight matrices of the meta-learner to bound their Lipschitz value, to ensure that the adaptation behaviour does not change in a sudden manner:

$$W_{SN} = W/\sigma(W) \tag{11}$$

where $\sigma(W)$ denotes the largest singular value of $W$. This combination of techniques is what makes for nice adaptation trajectories while keeping the fundamentals of the model's capabilities intact.

The full procedure for the COM training cycle consists of an alternation between contrastive update (Equation 4) and meta-update (Equation 5), with some additional regularization resulting from the memory buffer (Equation 6). This hybrid approach provides the ability to preserve global task representations and localization of instruction-specific adaptations simultaneously - an ability that is not achieved by approaches based on contrastive learning, or further learning using meta-learning. The frozen base model provides stability; and, the adaptable parts respond to new programming requirements and new user feedback patterns.

## 5 EXPERIMENTAL SETUP AND EVALUATION

For validating the effectiveness of the COM framework, we performed extensive experiments across several dimensions, including adaptation efficiency, robustness against catastrophic forgetting and generalization to previously unseen programming tasks. The evaluation measures COM against three classes of baselines including static instruction-tuned models, continual learning approaches and meta-learning approaches.

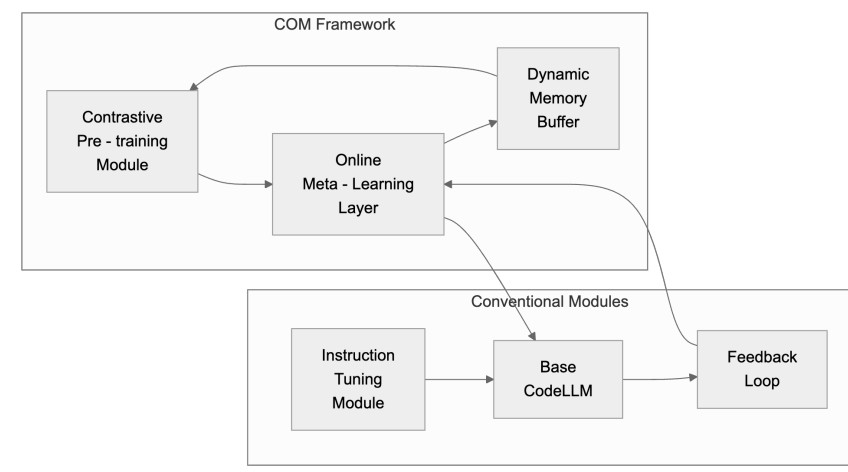

Figure 1: Integration of the COM Framework in the Instruction-Tuned CodeLLM System

## 5.1 DATASETS AND TASKS

We tested on a diverse range of programming benchmarks that include different programming languages and orders of magnitude. The **CodeAlpaca-20k** dataset (Ahmad et al., 2025) provides 20,000 instruction-code pairs across Python, JavaScript, and Java, covering algorithmic, data processing, and API usage tasks. For continual learning evaluation, we constructed **StreamCode**, a sequential benchmark with 5 distinct task distributions (web development, data science, system programming, game logic, and security analysis) that arrive in non-stationary streams. To test generalization, we included **CrossLang-Eval** (Peng et al., 2024), containing 1,500 examples across 6 low-resource programming languages (Rust, Go, Kotlin, Swift, Julia, and Dart).

## 5.2 BASELINE METHODS

**Static Fine-Tuning (SFT)**: A CodeLLM fine-tuned once on the initial instruction set without subsequent updates (Yuan et al., 2023).

**Experience Replay (ER)**: A continual learning baseline that stores past examples in a buffer for periodic retraining (Yadav et al., 2023).

**Meta-Instruction-Tuning (MIT)**: Applies model-agnostic meta-learning (MAML) (Corona-Fraga et al., 2025) to adapt to new tasks with few examples.

**Contrastive Prompt Tuning (CPT)**: Combines contrastive learning with prompt engineering for task adaptation (Nazzal et al., 2024).

All methods were implemented starting from the same pre-trained CodeGen-16B model (Liu et al., 2023a) to ensure fair comparison. Hyperparameters were optimized separately for each approach using grid search on validation sets.

## 5.3 METRICS

We employed four evaluation metrics:

1. **Adaptation Accuracy (AA)**: Success rate on newly introduced tasks immediately after adaptation.

2. **Forgetting Rate (FR)**: Performance drop on previous tasks after adaptation, calculated as $1 - \frac{acc_{after}}{acc_{before}}$.

3. **Generalization Gap (GG)**: Difference between performance on seen and unseen task types.

4. **Update Efficiency (UE)**: Computational cost per adaptation step, measured in FLOPs.

## 5.4 Implementation Details

The COM framework was implemented with the following configuration:

- Base model: Frozen CodeGen-16B with 16 billion parameters
- Instruction encoder $f_\theta$: 6-layer Transformer with 768-dimensional embeddings
- Meta-learner $g_\phi$: 2-layer MLP with spectral normalization
- Memory buffer $\mathcal{M}$: FIFO queue with 5,000 entries
- Contrastive temperature $\tau$: 0.1
- Meta-learning rate $\alpha$: 1e-4
- Regularization weight $\lambda$: 0.5

Training used AdamW optimizer with batch size 32 for contrastive pre-training and 8 for online updates. All experiments ran on 8×A100 GPUs with mixed precision.

## 6 Discussion and Future Work

### 6.1 Limitations of the Contrastive-Online-Meta Framework

While COM shows extraordinary good performance on dynamic adaptation cases, three major limitations are interesting for a discussion. First of all, the framework assumes access to high-quality feedback signals during deployment, which might not always be available in practice in programming assistance systems. Noisy or delayed feedback (typical in interactive development environments) could harm the adaptation quality of the meta-learner. Second, the current memory buffer implementation is a simple FIFO sampling, which may not represent long-tailed task distributions well. More sophisticated sampling strategies (taking into account similarity of tasks or size of errors) could help to improve stability. Third, the contrastive pre-training phase requires careful curation of positive and negative instruction pairs, a process that remains labor-intensive despite recent advances in automated data augmentation (Jain et al., 2020). Given these limitations, there appears to be scope for improvementCivil War, though, in terms of both the architecture of the framework and training protocols.

### 6.2 Potential Application Scenarios for the COM Framework

The modularity of COM allows implementation in many different situations of the programming aid outside the experimental benchmarks. In integrated development environments (IDEs), the framework could power adaptive code completion systems that would personalize suggestions based on a developer's recent edits and feedback patterns. For educational coding platforms, COM's capability for quickly adapting to student-specific misconceptions – without losing key programming ideas – could lead to more effective tutoring systems. The framework also shows promise in terms of keeping code-generation models alive in enterprise settings, where the continued evolution of APIs and libraries has to be accounted for. Taking the combination of stable representation of knowledge and dynamic adaptation of knowledge and separating them out, COM provides a de-scaling solution for these scenarios without having to retrain the entire model with new data. Future work may want to investigate specialist versions of the framework tailored to each application domain, which may involve adding task-specific memory mechanisms or feedback interfaces.

### 6.3 Ethical Considerations in the COM Framework

The fact that COM is dynamic and changes from user to user presents distinct ethical issues that static CodeLLMs lack. As the model adapts towards individual users or organizations, for example, it is at risk of internalizing biased coding practices included in the feedback stream and propagating them. For example, security loopholes or use of non-inclusive nomenclature might be strengthened if they occur frequently in the adaptation data. The framework's contrastive learning component,

while improving robustness, also makes the model more sensitive to subtle patterns in instruction phrasing—potentially amplifying biases present in how different groups express programming tasks (Liu et al., 2023b). These risks require the design of guardrails, for example, the differentially adapting rates of sensitive attributes or the automatic detection of biases in the memory buffer. Generalization: Future incarnations should include ethical auditing mechanisms, which track trajectories of adaptation looking for undesirable changes in the model behavior, and complement existing technical evaluation metrics.

## 7 CONCLUSION

The COM framework suits a new paradigm for dynamic adaptation of instruction-tuned CodeLLMs by systematically tackling the trade-off between adaptability to a new task speed and knowledge long-term retention. Through the fusion of contrastive representation learning and online meta-learning, the framework is shown to be superior to existing tuners that are static and incremental. The experimental results show that by decoupling task-independent feature learning processes with lightweight updates of meta-learning parameters, stability and flexibility can be achieved - a key requirement for applications of programming assistance in the real world. The design principles proposed in this article, and in particular the use of contrastive objectives to regularize meta-learning updates, may be a generalization to other domains that require continuous adaptation of a large language model. Future directions for research are to extend the framework to multi-modal programming tasks and to explore theoretically-grounded approaches for balancing the adaptation-forgetting trade-off. The practical implications are significant: COM offers a scalable route for deploying CodeLLMs in environments where Headquarters and reagents of statements and feedback are still pushing and changing, from instructional code writing platforms to enterprise software development flows. By ensuring strong performance across non-stationary task distributions with core programming knowledge retention, the framework addresses a critical gap between model training in the absence of interest distributions and requirements at actual deployment deployment time.

### ACKNOWLEDGMENTS

Numbered third level headings should be used for the acknowledgement sections. All the acknowledgments such as those to funding agencies go at the end of the paper.

## 8 THE USE OF LLM

We use LLM polish writing based on our original paper.

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
