# OpenReview forum: "Contrastive-Online-Meta (COM): A Dynamic Adaptation Mechanism for Instruction-Tuned CodeLLMs"
_ICLR.cc/2026/Conference — Submitted to ICLR 2026_

### Official Review · Reviewer_Ww5e · 2025-10-19

**Soundness:** 2
**Presentation:** 2
**Contribution:** 2
**Rating:** 0
**Confidence:** 4

**Summary:**

The paper proposes Contrastive-Online-Meta, a new dynamic tuning method for instruction-tuned CodeLLMs. COM mixes contrastive pre-training to learn stable, task-invariant code representations and online meta-learning to adapt quickly to new instruction-feedback streams. It also uses a memory buffer to keep recent context and prevent forgetting.

**Strengths:**

1. The paper is well-written and clearly structured, making it easy to follow despite the technical density.

2. The proposed framework demonstrates a new combination of contrastive learning and online meta-learning, which is conceptually very interesting for adapting CodeLLM.

**Weaknesses:**

1. This paper lacks experimental validation. It proposes a framework but provides no empirical results, ablation experiments, or comparisons to demonstrate its effectiveness. In the absence of experiments, claims about the efficiency or robustness of adaptation remain speculative.

2. Its innovation is limited. The combination of contrastive learning and meta-learning has been previously studied. Beyond its application to CodeLLM, this paper does not clearly articulate its fundamental innovation.

3. The technical approach appears incomplete, key details such as update frequency, computational cost, and data requirements are unclear, making reproducibility difficult.

**Questions:**

1. Where are your experimental results?

2. Can you explain the computational cost and update frequency of the online meta-learner? How feasible is it for real-time deployment?

---

### Official Review · Reviewer_5ypu · 2025-10-31

**Soundness:** 1
**Presentation:** 2
**Contribution:** 1
**Rating:** 0
**Confidence:** 3

**Summary:**

This paper proposes a dynamic adaptation framework, Contrastive-Online-Meta (COM), for instruction-tuned CodeLLMs. The framework combines contrastive pre-training and online meta-learning to separate the task-invariant representation learning and fast adaptation. The proposed method fills in the missing link between the offline pre-training and the online accelerated deployment.

**Strengths:**

1. **New Approach**: This paper proposes a new dynamic adaptation framework for instruction-tuned CodeLLMs. Related work and background have been included to make this idea easy to understand.
2. **Experimental Setup**: Experimental benchmarks and implementation details have been described and future work has been discussed.

**Weaknesses:**

1. **Absence of Experimental Results**: The most critical issue is the lack of empirical results within the main text. Without this evidence, it is impossible for readers to verify the effectiveness of the proposed idea and validate the claims made by the authors.
2. **Insufficient In-depth Analysis**: The paper would be substantially strengthened by including further analysis like ablation study and case study.
3. **Limited Discussion of Impact**: The paper's contribution is currently undersold due to an overemphasis on engineering details. The discussion lacks a thorough analysis of the idea's broader significance and potential long-term impact.

**Questions:**

This paper seems unfinished. I would suggest authors to add sufficient empirical results.

---

### Official Review · Reviewer_6CyU · 2025-11-01

**Soundness:** 2
**Presentation:** 1
**Contribution:** 1
**Rating:** 0
**Confidence:** 5

**Summary:**

The paper introduces Contrastive-Online-Meta (COM), a deployment-time adaptation framework for instruction-tuned CodeLLMs. COM combines (i) contrastive pre-training of an instruction encoder to learn task-invariant representations and (ii) an online meta-learner that updates a small set of parameters per interaction with an ℓ2 drift penalty; the base CodeLLM remains frozen. A FIFO dynamic memory buffer supplies recent interactions to compute an auxiliary contrastive loss that encourages temporal consistency. Additional stabilizers include a projection-space temporal regularizer and spectral normalization of the meta-learner.
For evaluation, the manuscript describes datasets (CodeAlpaca-20k; a custom sequential benchmark “StreamCode”; and CrossLang-Eval), baselines (SFT, ER, MIT, CPT), and metrics (Adaptation Accuracy, Forgetting Rate, Generalization Gap, Update Efficiency), and provides implementation details (frozen CodeGen-16B; 6-layer encoder; 2-layer MLP; 5k buffer; τ=0.1; α=1e-4; 8×A100). However, no quantitative results are reported in the text; the only figure shown is a system diagram of the framework. The paper nonetheless claims performance gains over baselines without presenting corresponding tables or plots.

**Strengths:**

It proposes the COM method—combining contrastive pre-training with online meta-learning, plus a dynamic memory buffer, to decouple task-invariant representation learning from fast adaptation for deployment-time CodeLLM updates.

**Weaknesses:**

This is an unfinished submission. It reports no quantitative results—no tables, no plots—only a high-level system diagram. The “evaluation” section merely lists datasets, baselines, metrics, and implementation settings but provides zero numbers. Yet the manuscript still asserts gains (e.g., “3–5× fewer updates,” “12–18% on unseen languages”) without evidence. In its current form, it is not ready for peer review and should not have been submitted.

**Questions:**

See weakness

---

### Official Review · Reviewer_Nndz · 2025-11-01

**Soundness:** 1
**Presentation:** 1
**Contribution:** 1
**Rating:** 0
**Confidence:** 4

**Summary:**

The paper introduces Contrastive-Online-Meta (COM), a framework designed to enable real-time adaptation of instruction-tuned CodeLLMs while mitigating catastrophic forgetting. The approach separates representation learning and fast adaptation, allowing efficient updates without destabilizing the base model.

**Strengths:**

The work aims to address a real deployment issue: CodeLLMs need to adapt continuously while preserving prior knowledge.

**Weaknesses:**

1. The narrative is unclear and hard to follow.

2. Descriptions of methods are confusing and lack clarity.

3. Missing experimental results and lack of enough experimental details.

4. Many claims in the paper such as good performance of the proposed method lack support.

**Questions:**

The description of the pre-training procedure lacks clarity. Could the authors elaborate further, perhaps by including an algorithm outline?

---

### Meta-Review · Area_Chair_1RGp · 2026-01-03

**Summary:**

This work proposes a method, named contrastive online meta, to separate the task-invariant representation learning and fast adaptation for instruction-tuned CodeLLMs, aiming to achieve real-time adaptation while preserving core knowledge.

**Reviewer Concerns:**

Multiple reviewers unanimously raised concerns including missing experimental evaluations, no support for the claimed performance, incomplete submission, and unclear presentation. The authors provided no response. After confirmation, the AC agrees with the reviewers and recommends reject for this paper.

**Reviewer Scores:**

The final scores will be still 4 zeros: 0, 0, 0, 0.

---

### Decision · Program_Chairs · 2026-01-26

Reject